# Algorithmic Approach to Virtual Machine Migration in Cloud Computing with Updated SESA Algorithm

**DOI:** 10.3390/s23136117

**Published:** 2023-07-03

**Authors:** Amandeep Kaur, Saurabh Kumar, Deepali Gupta, Yasir Hamid, Monia Hamdi, Amel Ksibi, Hela Elmannai, Shilpa Saini

**Affiliations:** 1Chitkara University Institute of Engineering and Technology, Chitkara University, Punjab 140401, India; 2Information Security and Engineering Technology, Abu Dhabi Polytechnic, Abu Dhabi 111499, United Arab Emirates; 3Department of Information Technology, College of Computer and Information Sciences, Princess Nourah bint Abdulrahman University, P.O. Box 84428, Riyadh 11671, Saudi Arabia; 4Department of Information Systems, College of Computer and Information Sciences, Princess Nourah bint Abdulrahman University, P.O. Box 84428, Riyadh 11671, Saudi Arabia; 5Department of CSE, Chandigarh University, Mohali 140413, India

**Keywords:** cloud computing, virtual machine, migration, power consumption, SESA

## Abstract

Cloud computing plays an important role in every IT sector. Many tech giants such as Google, Microsoft, and Facebook as deploying their data centres around the world to provide computation and storage services. The customers either submit their job directly or they take the help of the brokers for the submission of the jobs to the cloud centres. The preliminary aim is to reduce the overall power consumption which was ignored in the early days of cloud development. This was due to the performance expectations from cloud servers as they were supposed to provide all the services through their services layers IaaS, PaaS, and SaaS. As time passed and researchers came up with new terminologies and algorithmic architecture for the reduction of power consumption and sustainability, other algorithmic anarchies were also introduced, such as statistical oriented learning and bioinspired algorithms. In this paper, an indepth focus has been done on multiple approaches for migration among virtual machines and find out various issues among existing approaches. The proposed work utilizes elastic scheduling inspired by the smart elastic scheduling algorithm (SESA) to develop a more energy-efficient VM allocation and migration algorithm. The proposed work uses cosine similarity and bandwidth utilization as additional utilities to improve the current performance in terms of QoS. The proposed work is evaluated for overall power consumption and service level agreement violation (SLA-V) and is compared with related state of art techniques. A proposed algorithm is also presented in order to solve problems found during the survey.

## 1. Introduction

Cloud computing is one of the most emerging fields in modern-day developments. A cloud network is comprised of three layers of services as follows:(a)Infrastructure as a service (IaaS);(b)Platform as a service (PaaS);(c)Software as a service (SaaS).

To speed up the computation efficiency, the physical machines (PMs) (machines with physical attributes) are supported with virtual machines (VMs). There are two issues in the association of VM to the PM.
(a)Allocation of a new VM to the PM;(b)Management of existing allocated VMs.

Identify the applicable funding agency here. If none, delete this text box.

Both processes consume power and, hence, if they are not managed well, high power consumption will be observed. The world is already suffering from global warming and, hence, in such a scenario, high power consumption is not affordable. From the records, it is analyzed that one data centre consumes as much energy as 25,000 households [1]. Energy consumption is integral power over a given unit of time and can be defined by Equation (1)
(1)$$E=\int_{t1}^{t2}Pc\;dt$$
where *Pc* is consumed power and *dt* is the time interval when the power is consumed which can satisfy the quality of service (QoS) requirements of users considering the service level agreement (SLA). An energy-efficient cloud model is also termed a green cloud. Green cloud architecture has four elements, as shown in Figure 1.
(a)Consumer or broker: A consumer or customer of the cloud submits its requirement directly to the cloud or gets it submitted by a broker;(b)Cloud-service allocator (CSA): The cloud infra is not directly associated with the user and, hence, the CSA negotiates on SLA, prices, and other terms between the service provider and the customer. The service allocation has a service scheduler associated with a CSA, which deals with the completion time and scalability of customer demand;(c)Physical machine (PM);(d)Virtual machine (VM).

VM allocation has two scenarios, namely the admission of a new VM and migrations of one VM to another PM. The first scenario is observed at the preliminary stage of the cloud server initialization. The most basic architecture was proposed by Dr. Rajkumar Buyya in 2010, which is named modified best fit decreasing utilization (MBFD) which is an extension of the best fit decreasing algorithm (BFD). BFD was developed by Baker et al. in the late 1980s and was used further in the early stages of cloud and grid computing [2]. When a VM is to be migrated from one PM and is to be allocated to another PM, two issues were addressed.
(a)Hotspot detection: Which PM is to be detected for the migration?(b)Destination PM detection: Where to migrate?

The MBFD algorithm uses CPU utilization in order to allocate the VMs at the preliminary stage, i.e., when a new VM is to be allocated to the PM. The MBFD algorithm sorts the CPU utilization in descending order and looks for the resources with the highest CPU utilization containing VM. The algorithm also checks the most feasible host with the least power consumption. The pseudocode for MBFD is in Algorithm 1 as follows.
**Algorithms 1:** MBFD*Inputs: VM Requirements and Specifications, Host Specifications* *(a)*  *Sort all VM as per the CPU Utilization in descending order* *(b)*  *For every VM in the VM List(Sorted)* *(c)*  *Check if Host can satisfy the VM needs or not* *(d)*  *Calculate Pc* *(e)*  *Check if Pc is least* *(f)*  *If Yes, Allocate VM to Host* *(g)*  *Reduce Host resources by the amount which is consumed by the VM* *(h)*  *Pick Next VM*

As time passed on, amendments have been monitored in MBFD. In 2015 Lu. X and Zhang [3] presented an enhanced MBFD algorithm for allocation, which considered load over the PMs. Lu and Zhang neutralized bandwidth, random access memory (RAM), and CPU utilization as follows. (2)$$\begin{array}{l} \mathrm{Normalized}\mathrm{Demand}\mathrm{(ND)}=\\ \;w1\times Ram+w2\times CPU+w3\times Bandwidth \end{array}$$
where w1, w2, and w3 are weights for RAM, CPU, and bandwidth of the VM.

The modified MBFD algorithm follows MBFD by replacing the CPU utilization by ND. It also adds up a hot migration policy in which a threshold is set considering all the loads over the PMs and, if the PM exceeds the threshold, then VMs are selected for the migration. The load calculation is done by applying three conditions, as given in Equation (3). The equations considered host CPU utilization (HCPU), average CPU utilization (ACPU), host RAM utilization (HRAMU), average RAM utilization (ARAMU), and host bandwidth utilization (HBU), along with average bandwidth utilization (ABU) for overloaded migrations.
(3)$$\begin{array}{l} 0\;Otherwise\\ \;1\;if\;HCPU\;>\;ACPU||HRAMU>ARAMU||HBU\;>\;ABU \end{array}$$

Considering load as an effective parameter, Mann et al. presented a computation core-aware load-based MBFD algorithm, which also considered the CPU cores involved in the allocation process. In order to enhance power management, the distribution of load is minutely analysed over the individual cores, and core-to-core migration is also incorporated [4]. The usage of natural computing is also observed in MBFD advancements. As per “de Castro et al.”, natural computing is a terminology which encompasses the algorithmic architecture which takes inspiration from nature or that employs natural phenomenon to complete a given task [5]. Cloud computing uses this phenomenon in order to identify the PM whose VMs are to be migrated. Kansal et al. [6] used swarm intelligence (SI) for VM selection (swarm intelligence)-based behaviours observed in groups. For example, the behaviour of ants, the behaviour of cuckoos and fireflies, etc.). The PM from where the VMs are to be migrated is termed an active source node. The algorithmic architecture is further extended to use an artificial bee colony (ABC) for the selection of the destination node where the VMs are to be migrated. The source node is selected by the computation of the “Attraction Index (AI)”, which is computed by the evaluation of “Consumed Energy (CE)” as shown in Equation (4).
(4)$$CE=\frac{\left(\sum_{i=1}^{n}\sum_{j=1}^{k}CPU_{ij}\right)\left(\left(\sum_{i=1}^{n}\sum_{j=1}^{k}MU_{ij}\right)\right)}{M}$$
where *n* is the total amount of VM executing over PM and *k* denotes the total amount of jobs allocated to *i*th VM. *M* corresponds to the total amount of memory units executing on the *i*th VM. In addition to CE, the computation time of the node or *NCT* corresponds to the alive node to compute the total invested time in order to complete supplied jobs is also calculated. NCT formulation is given by Equation (5).
(5)$$NCT_{active}=\sum_{i=1}^{n}\sum_{j=1}^{k}NCT_{ijactive}$$
where $NCT_{ij}$ is the execution time for *k* number of jobs supplied to *n* number of VMs. Using *CE* and *NCT*, *AI* is calculated by Equation (6).
(6)$$AI_{i}=AI_{active}(CE_{active},NCT_{active})$$

By sorting the *AI* in decreasing order, the lowest *AI* index will be selected as the PM from which the VMs are to be migrated. In order to choose the VMs to be migrated, the load on each VM is calculated by using Equation (7).
(7)$$Load_{ij}=\frac{\sum_{i=1}^{n}job_{i}}{\int_{0}^{t}Pc\;dt}$$

VMs with a high load compared to the average load are considered to be migrated from the active PM. Durgadevi et al. [7] also used swarm-based hybridization of leapfrog search and cuckoo search for the VM and PM selection for migration. In addition to CPU utilization, load, and computation time, the authors introduced entropy for the computation of VMs to be migrated. 

When it comes to selecting the VMs for the migration, several research articles have focused on VM selection that is based on threshold [1,3,6,7,8]. In order to detect the hotspot, Beloglazov et al. proposed the minimization of migrations (MM). MM has two thresholds, namely upper and lower. If the usage of the host CPU is lower compared to the lower threshold, all host VMs are migrated and if the utilization of the host CPU is higher than the upper threshold, some VMs are migrated through the PM. The working is illustrated in Algorithm 2.
**Algorithm 2:** MM AlgorithmInput: Hotlist Output: Migration List  1. Repeat until the hUtil is greater than upper threshold  2. Forevery vm in the Host’s vmList  3. If vm.utilization() > hUtil − upper threshold  4. Evaluate t as t ← vm.utilization() − hUtil + upper threshold  5. Keep vm as best fit vm until greater t is not attained or host’s utilization does not go below upper threshold  6. Reduce hUtil by best fit vm.utilization()  7. Add vm to migration list 

In this work, the main issue of power consumption, SLA-V, and optimal resource utilization has been addressed by allocating VM and migrating to the best available host in an energy-efficient manner. It has been done by developing an energy-efficient model using a modified SESA algorithm in order to improve the efficacy of the proposed model. Two additional parameters such as cosine similarity and bandwidth utilization have been employed.

## 2. Literature Survey

Masdari and Khezri (2020) introduced a detailed analysis of the process of VM migration [9]. Dubey and Sharma (2020) worked for the betterment of the utilization of the resources by introducing intelligent water drop (IWD) for the selection of the VMs to be migrated [10]. To check the effectiveness of the proposed algorithm, the authors increased the load through load files which are globally available on cloudbus.org (accessed on 24 June 2023).

Joshi and Munisamy (2020) introduced the dynamic degree balance (DDB) in collaboration with MBFD. The authors also considered the load mechanism in contrast to the VM allocation mechanism and, hence, the two policies run simultaneously to point the VM to an existing PM. The authors compared the algorithm with two static algorithms such as shortest job first (SJF) as well as first come first serve (FCFS). The rate of imbalance and waiting time to experiment have been considered as evaluation parameters and work simulation has been done by CloudSim [11]. 

Ruan et al. (2019) described a model for allocating VMs including migrations that leverage the PPR in different kinds of hosts. In comparison to three simple energy-efficient algorithms for allocating and selecting VMs, ThrRs, MadMmt, and IqrMc, the experimental performance showed that the system could reduce energy consumption to 69.31% for many types of host computers, including certain shutdown speeds and migration times along with a minor degradation in performance for cloud data centres [12,13].

Jin et al. (2019) implemented a VM allocation in a cloud data centre with a speed switch and VM consolidation, and the energy efficiency and response results were considered by the data centre. Consequently, to study the queuing model, the authors set up a contiguous multiple-server queuing model [13].

Jia et al. (2019) proposed the latest VM allocation approach for power consumption and load balancing to resolve security issues [14]. 

Nashaat et al. (2019) developed an updated smart elastic scheduling algorithm (SESA) to cluster the VMs to be migrated. SESA aimed to obtain a load-balanced allocation and migration policy. SESA used CPU utilization and RAM as the parameter for the VMs to be located on the same physical machine [8]. 

Gamal et al. (2019) consider load as a primary parameter for VM allocation and migration. In order to migrate the VMs, a bioinspired VM migration policy was presented, which uses RAM and CPU utilization as the main parameters [15].

Basu S. et al. (2019) have resolved the problem of workload using nature-inspired GA along with the improved scheduling of VMs. Every chromosome of GA has been defined as a node, and the VM is allocated to a node related to the chromosome’s genes. After that, based on the crossover and mutation operator, the allocation of VM has been performed. The results have indicated that the proposed approach has performed well in terms of resource utilization and load balancing. However, the area of cloud computing and VM allocation is not new. However, still, the proposed algorithm presents a new behaviour of the cuckoo search algorithm for the selection of VMs to be migrated. The selected VMs are cross-validated using a feed-forward back propagation model which dual checks the allocation, which further ensures optimal power consumption [16].

Zhang et al. (2019) have presented a new, as well as effective, evolutionary-based strategy to allocate VM and thereafter minimized the energy consumption of CDC. The comparative analysis through performing simulation on CloudSim has demonstrated that the proposed method enables for rapidly implementing an allocation corresponding to a set of VMs in an optimized way and integrating more VMs with fewer PMs. Higher energy efficiency than baseline energy efficiency methods has been achieved. Specifically, compared with the most advanced methods, the complete profit gain as well as saves consumed energy in a preset approach up to 24% and 41%, respectively. In addition, our approach can enable CDC to satisfy more end-user requests [17].

Jana et al. (2019) have presented a Modified PSO method in which the researchers have considered two parameters, namely average scheduling length along with the rate to execute effectively. The comparison with baseline techniques such as Max–Min, Min–Min, and simple PSO approaches have been performed to analyse the effectiveness of the proposed work. The main motive of this paper is to allocate resources by properly managing the workload in a cloud server, which is performed by using a task scheduling algorithm [18].

Gawali and Shinde (2018) have used a metaheuristic approach by integrating the two optimization approaches, namely, bandwidth-aware divisible scheduling with BAR optimization for task scheduling along with solving the purposes of resource allocation. Using the modified analytic hierarchy process (MAHP), each task is processed. The experiments have been performed on a cyber snake in which epigenomics is passed as an input task. The performance in terms of turnaround time and response time has been compared to the BATS and IDEA approaches. The proposed approach found better results with high resource utility [19].

Verma and Kaushal (2017) have proposed an integrated approach and given the name of “Hybrid Particle Swarm” optimization (HPSO). The algorithm is used to maintain the scheduling workflow problem using a fitness function designed for an HPSO approach and used in an IaaS cloud. The optimization technique has been integrated into the deadline constraint heterogeneous earliest finish time (HEFT) algorithm [20].

Wang et al. (2017) presented an approach on the basis of classifying memory pages into five categories with the aim to minimize total data transferred, by which total migration time has also been reduced. Such categories are inode, cache, anonymous, free memory pages, and kernel, by which they transmit only the first three categories of memory pages since they are required for the execution of the kernel. However, normal execution has not affected by free memory pages, avoiding the transfer of cache memory pages leads to degradation of performance because of inconsistency among actual transmitted memory pages along with the kernel state. The attained findings contrast with the default per copy of KVM and indicated that this scheme can minimize average migration time by 72% [21].

Kansal et al. (2016) proposed a firefly algorithm-based approach in which the maximum loaded VMs are migrated to the least loaded PM without violating the performance of energy efficiency in CDC. From experiments, the improvement in energy consumption of 44.39%, minimization in migration of 72.34%, and hosts of up to 34.36% have been saved [6].

Akbar et al. (2016) introduced a recent task scheduling approach named median deviation-based task scheduling (MDTS) that utilized median absolute deviation (MAD) of the expected time to compute (ETC) in a task as a key attribute to determine ranks of that task. Authors utilized a coefficient-of-variation (COV)-based approach that considered task along with device heterogeneity with the aim to get the ETC of specific DAG. The execution in the cloud program can be visualized as a collection of various tasks depicted through DAG which executes in their logical sequence. To accomplish the improved performance along with enhanced efficiency in a cloud environment, the prioritization of such tasks plays a key role. The evaluation of presented schemes has been done in different circumstances through DAGs with real-world applications. The outcome indicated that the MDTS approach produced high-quality schedulers along with significantly minimising the makespan by 25.01% [22].

Esa and Yousif (2016) introduced a novel job-scheduling mechanism that uses the firefly algorithm with the aim to minimize the time to execute jobs. The presented frameworks for a job, along with resource information such as the length of the resource, along with job speed identifiers. In the work-schedule process, the proposed scheduling function first generates a collection of jobs and resources to produce the population by assigning the work to the resources randomly and measuring the population using fitness values that reflect the execution time of work. Next, to provide the shortest execution time for the job, the function consumed iterations to reproduce the population on the basis of the actions of the fireflies to deliver the best job plan. The Java language and the CloudSim simulator are used to implement various scenarios [23].

R. Durga Lakshmi (2016) proposed a minor modification in the working architecture of the genetic algorithm (GA) to optimize the overall system efficiency. Quality of service (QoS) management distributes the problem generated by cloud applications. Tools were addressed to guarantee service levels in terms of performance, reliability, availability and so on. The system’s waiting time should be reduced to boost the QoS in the system. GA is a technique of heuristic search that provides the optimal solution for the task. The discussed approach produces a scheduling algorithm based on GA to maximize the system’s overall latency. The cloud environment is split into two parts: the cloud user (CU) is one and the cloud-service provider (CSP) is another. A service request is sent to the CSP by the CU and all the requests have been stored in the request. The queue (RQ) explicitly interacts within the C with the GA module queue sequencer (GAQS) [24].

Deshpande et al. (2016) suggested a combining copying technique for live VM migration. The amount of transferred memory pages has decreased by this technique. The hidden pages for dirtied work can be demanded by the destination or actually printed through the source [25].

Forsman et al. (2015) have presented two schemes to cooperate for automated live migration among multiple VMs. The researchers have used a push scheme in order to migrate VMs from overload PMs to under-loaded PMs. Using this strategy, the under-utilized PMs have requested to the heavy-workload servers that the load can be passed to under-utilized servers. Using three conditions, namely, PM state after migration, cost of migration, and workload distribution, at what time VM should be migrated was determined. The redistribution of workload and the quick attainment of a balanced cloud system has been the key objective of this work, whereas, how SLA violation has been minimized was not explained in this paper [26].

M. S. Pilavare et al. (2015) have presented a number of schemes to enhance the existing load-balancing algorithms in CDC. Among all schemes, the researchers have stated that GA is superior to various technologies. GA uses a randomly selected processor as the input and then processes it. It was thought that the same priority was given to processors and jobs but this is not the case. The logarithmic matrix with the least squares is provided here to increase the efficiency of the GA. When a PM is not performing anything, ethically, it is termed as idle, which wastes the available resources and, hence, to solve the problems of idleness through observation, the proposed algorithm performed better. It has been concluded by observing the method of the GA randomly selected processors and concluded that the processors with a higher fitness value have been utilized and that the virtual machines with lower fitness values have left. The simulation has been done using a cloud simulator [27]. 

Garg et al. (2014) have resolved the problem of resource allocation in CDC in which the task has been performed with different workload conditions using distinct applications. An admission control, along with a scheduling mechanism, has been proposed which can not only maximize the use of resources but also guaranteed that the end-user QoS requirements must be satisfied along with the SLA parameters. From the research, it has been concluded that the understanding of different kinds of SLAs and the applicable penalties along with the workload mix are very important for better resource allocation and utilization of CDC [28]. 

Song et al. (2014) presented a method of resource allocation as an online box problem. To dynamically assign available resources in the cloud data centre according to the application requirements and maximize the number of active servers to help green computers, virtualization technology has been used. To resolve this problem, a variable item size boxing (VISBP) algorithm has been presented that has been implemented carefully using both VM and PM as classification functions. Small variation has been tolerated as long as the classification rules have been satisfied by the system. The finding indicated that the performance of VISBP is better for migration of a hotspot as well as in load balancing, in contrast to the existing algorithms. The key difficulty in its implementation may be the assumption that all PMs in this work have homogeneous unit power [29].

Beloglazov et al. (2012) investigated the research on energy-saving computing and put forward: (i) cloud energy-saving management architecture principles; (ii) energy-saving resource allocation strategies and scheduling algorithms that consider the QoS expectations and power user features of devices; and (iii) deal with some openness research challenges, such challenges may offer immense advantages for suppliers of resources and customers. By performing a performance assessment using the CloudSim toolkit, the methodology was validated. The findings showed that the model of cloud computing has great potential because it can save a lot of costs and has great potential in complex workload scenarios to increase energy efficiency [1].

Quang-Hung et al. (2013) proposed a genetic algorithm for VM allocation. For experiments, the workload in a computer lab at a university for one day has been examined. The VMs are sorted in start time using the BFD algorithm. The results obtained using the proposed GAPA scheme have been compared with the existing approach and find that the proposed algorithm achieves less energy consumption. The results show that the computation time has been reduced to a great extent [30]. 

Madhusudhan et al. (2013) introduced an energy-aware VM placement strategy in combination with swarm-inspired particle swarm optimization (PSO) algorithm to minimize the total energy consumption rate and hence enhances server usage and provide satisfaction to mobile cloud users. The parameters, as well as the operators, have been redefined to resolve the discrete optimization issues. The parameters have been readjusted as the traditional PSO technique has failed to solve the problem of VM placement since it has been mostly utilized for solving only continuous optimization complexities. The PSO algorithm helps to minimize the searching time and, hence, saves energy with better server utilization [31].

Talwani and Singla (2021) suggested using an extended artificial bee colony (E-ABC) method. Results from simulations show that the E-ABC approach has high scalability [32,33,34]. The E-ABC strategy saved 15–17% more energy and resulted in 10% fewer migrations than the current concept [35]. The proposed approach has a higher computation time.

Dia et al. (2022) studied a cloud-assisted fog computing framework with task offloading and service caching presented to ensure effective task processing. The framework allowed tasks to make offloading decisions for local processing, fog processing, and cloud processing, with the objective of minimizing task delay and energy consumption, taking into account dynamic service caching. To achieve this, a distributed task offloading algorithm based on noncooperative game theory was proposed. Furthermore, the 0–1 knapsack method was employed to realize dynamic service caching. Adjustments were made to the offloading decisions for tasks that were offloaded to the fog server but lacked caching service support [36].

Tran et al. (2022) addressed the challenges related to the migration of machines in cloud-computing architecture. The authors proposed a VM migration algorithm based on the concept of Q-learning and Markov decision-making models. The work comprised a training phase followed by the extraction phase. The superiority of the proposed algorithm in terms of feasibility and strengths of the extraction phase is demonstrated using comparative analysis against a max–min ant system, round robin and ant system algorithms [37].

Abedi et al. (2022) considered the heuristic method using the firefly algorithm and fuzzy approach to prioritize the tasks. The authors improved the population of the firefly algorithm with the intention of balancing the load, migration rate, average run time, and completion time. The authors used MATLAB software for implementation and to maintain the makespan to 50.16. The drawback of the study was that some objectives still need to achieve using better optimization [38].

Khan et al. (2022) presented a hybrid cuckoo search and particle swarm optimization (CU-PSO) approach for effective VM migration. The primary goals of this study are to shorten the duration of computations, decrease energy use, and lower the cost of relocation. The optimal use of available resources is another focus area. The study aim is supported by a simulation analysis that compares the efficiency of the hybrid optimization model to that of more traditional methods. The proposed approach is outperformed in terms of energy consumption, migration cost, resource availability, and computation time [39]. The proposed work is insufficient to address the issue of SLA violation.

Zhao, H. et al. (2023) suggested a VM performance-aware approach (PAVMM) using ant-colony optimization (ACO). The process continues by setting a goal of user-friendly VM performance optimization. The suggested work has goals for cloud-service vendors including reducing the overall migration cost and the number of active PMsThe proposed framework minimizes the energy consumption and minimizes the active host which improves the speed of operation but proposed work lacking in SLA violations [40].

Bali, et al. (2023) developed a priority-aware task scheduling (PaTS) algorithm specifically designed for sensor networks. The algorithm aims to schedule priority-aware tasks for data offloading on edge and cloud servers. The proposed sensor-network design includes dividing tasks into four distinct groups: very urgent (OVU), urgent (OU), moderate (OM), and nonurgent (ONU). To address this problem, a multiobjective function formulation was used and the efficiency of the algorithm was evaluated using the bio-inspired NSGA-2 technique. The results obtained demonstrated significant improvement compared to benchmark algorithms, highlighting the effectiveness of the proposed solution. Additionally, when the number of tasks was increased from 200 to 1000, subsequent improvements were observed. Specifically, the average queue delay, computation time, and energy showed overall improvements of 17.2%, 7.08%, and 11.4% respectively, for the 200-task scenario [41].

Singh et al. (2023) focused on the utilization of containerized environments, specifically docker, for big data applications with load balancing. A novel scheduling mechanism for containers in big-data applications was proposed, based on the docker swarm and the microservice architecture. Docker swarm was employed to effectively manage the workload and service discovery of big-data applications. The implementation of the proposed mechanism involved a case study that was initially deployed on a single server and then scaled to four instances. The master node, implemented using the NGINX service, ran the swarm commands, while the other three services were microservices dedicated to the implemented scenario for a big-data application. These three worker nodes consisted of the PHP front-end microservice, Python API microservice, and Redis cache microservice. The results of the study demonstrated that increasing workloads in big-data applications could be effectively managed by utilizing microservices in containerized environments, and docker swarm enabled efficient load balancing. Furthermore, applications developed using containerized microservices exhibited reduced average deployment time and improved continuous integration [42].

Kavitha et al. (2023) developed a novel approach called filter-based ensemble feature selection (FEFS) combined with a deep-learning model (DLM) for intrusion detection in cloud computing. The DLM utilized a recurrent neural network (RNN) and Tasmanian devil optimization (TDO) to determine the optimal weighting parameter. The initial phase involved collecting intrusion data from global datasets, namely KDDCup-99 and NSL-KDD, which were used for validating the proposed methodology. The collected database was then utilized for feature selection to enhance intrusion prediction. FEFS, a combination of filter, wrapper, and embedded algorithms, was employed for feature extraction, selecting essential features for the training process in the DLM. The proposed technique was implemented using MATLAB and its effectiveness was evaluated using performance metrics such as sensitivity, F-measure, precision, recall, and accuracy. The suggested strategy demonstrated significant improvements based on these performance metrics. A comparison was made between the proposed method and conventional techniques such as RNN, deep neural network (DNN), and RNN-genetic algorithm (RNN-GA) [43].

## 3. Motivation of the Research

This research work is inspired by the research done by Beloglazov et al. [1], Kansal et al. [6], and Nashaat et al. [8]. There is a possibility of enhancement in the SESA algorithm by adding a third variable, bandwidth utilization (BU), as done in [3]. In addition, the SESA can be improved by adding similarity measures other than ED. In addition to this, the allocation is done based on the clusters’ density only, which could be further improved.

## 4. Methodology

The methodology of the proposed algorithm is centric, as per the written objectives. The first objective is to analyse existing VM allocation and migration techniques in which the focus would be towards studying SESA, MBFD, and its other enhancement. The design of the proposed algorithm is divided into two segments, namely the organization of the clusters and the placement of VMs into the concerned clusters. Considering the VMs to be migrated, as per the MM algorithm, Nashaat et al. [8] in 2019 proposed a smart elastic scheduling algorithm (SESA) which clusters the VMs in order to migrate. Nashaat modified the k-means algorithm by using RAM and CPU utilization of the VMs and the host. In order to modify the k-means algorithm, Nashaat used Euclidean distance (ED).

### 4.1. Detection of the Hotspot

Detection of the hotspot, based on the unit threshold, which may be SLAV or load, is a common practice. Even if a PM is idle, it consumes 70% of its peak power and, hence, letting the PM sit idle would also be using the power without producing an output. A dual threshold policy which keeps one upper threshold and one lower threshold of utilization is observed to be practised commonly [11,12,13]. The upper and the lower threshold are responsible for the determination of the fact that whether all the VMs should be migrated from the PM or some VMs should be migrated.

### 4.2. Selection of VMs

The selection of VMs has been done by considering a lot of parameters such as utilization, time of migration, and correlation between the VMs [44,45]. Considering the essentials to reduce the power consumption, minimization of migrations (MM) and minimum utilization (MU) were proposed and have gained potential enhancements time by time [6,14].

### 4.3. Selection of PM

Once the VMs are selected, other than hotspot PMs, the rest of the PMs are listed down as per the required essentials and then the VMs are placed as per shown in Section 4.4. The listed PMs are now called the available PMs.

### 4.4. VM Migration

Putting the right VM to the right PM is the task of this section. The proposed CESCA algorithm enhances the k-means algorithm by using CPU utilization (CPU), bandwidth utilization (BU), and currently associated random access memory (AsRAM). The algorithm takes inspiration from Nashaat et al.’s clustering algorithm, the smart elastic scheduling algorithm (SESA) [8]. CESCA is subdivided into three parts for the calculation of the number of centroids and the identification of centroids, placement of VMs into the centroid, and prioritization of the created clusters. 

The resource assignment or allocation [46,47,48] has four steps for VM allocation, as demonstrated in Figure 2. The first step is the detection of the hotspot PM. The SESA algorithm given in Algorithm 3 finds the optimal number of clusters required for the VMs to be migrated. In order to excel in the performance, an association of VM to PM is done using the Euclidian distance.
**Algorithm 3:** SESAInput: hostList, VMList, Standard Deviation threshold, Output: high density arrangedcluster list of co-located VMs, allocation of VMs
 1. Find K points for selecting the optimal number of clusters
 2. Calculating K1 (for Parameter −> CPU)
 3. K1_maxpoint = hostlist.get_max(CPU)/VMList.get_min(CPU)
 4. K1_minpoint = hostlist.get_min(CPU)/VMList.get_max(CPU)
 5. K1 = average(K1_maxpoint,K1_minpoint)
 6. Calculating K2 (for Parameter −> RAM)
 7. K2_maxpoint = hostlist.get_max(RAM)/VMList.get_min(RAM)
 8. K2_minpoint = hostlist.get_min(RAM)/VMList.get_max(RAM)
 9. K2 = average(K2_maxpoint,K2_minpoint)
 10. K = average(K1, K2)
 11. Select the initial centroid as a pair of two values (CPU, RAM)
 12. centroids [1, 1] = Get_average_CPU(VMList)
 13. centroids[1, 2] = Get_average_RAM(VMList) ∗
 14. Find the remaining K − 1 centroids,
 15. for each mth centroid number do, Where m takes values from 1 to K − 1
 16. $Calculate\;the\;Euclidian\;distance\;\left(ED\right)$ between previous centroid and (CPU, RAM)
 17. parameters of each VM in VMList
 18. for each jthVM in VMList do,
 19. Where j takes values from 1 to no. of VMs in VMList
 20. Ecu_dis[j] = find_Eculedian(VMList.get(j), centroids[m, 1], centroids [m, 2]) ∗
 21. end for
 22. Choose the Next Centroid to be (CPU, RAM) values for VMwithmaximum ED 23. centroids[m+1,1] = VMList.get(get_index_forMaxValue(Ecu_dis)).get(CPU)
 24. centroids[m+1,2] = VMList.get(get_index_forMaxValue(Ecu_dis)).get(RAM)
 25. end for 26. Calculate the ED between each VMs and all Cluster’s centroids
 27. for each jthVM in VMList do, Where j takes values from 1 to no. of VMs in VMList
 28. for each mth centroid number do, Where mtakes values from 1 to K − 1
 29. ED[m] = find_Eculedian(VMList.get(j),centroids[m,1],centroids[m,2])∗
 30. end for
 31. Append VM to the Cluster with minimumED
 32. Cluster = Append_in_Cluster(get_index_forMinValue(ED), VMList.get(j))
 33. end for 34. Arrange the co-located VMs 35. for each ith VMs cluster list in cluster do 36. arrangeBy Co-locatedVMs(Cluster.get(i)) 37. end for 38. VMList = arrangeBy HighDensityCluster(Cluster)


### 4.5. VM Migraion


PSEUDO-CODE CESCAInputs: $H_{List},VM_{List}$ Output: prioritized//Calculate the total number of centroids P//Calculate Px,Py and Pz, Where Px is based on CPU, Py is calculated based on AsRAM
//and Pz are calculated based on BU
(8)$$\begin{array}{l} Px_{max}=\frac{H_{List}.getmax(CPU)}{VM_{List}.getmin(CPU)};\;Px_{min}=\frac{H_{List}.getmin(CPU)}{VM_{List}.getmax(CPU)}\\ Px=\frac{Px_{min}+{Px}_{max}}{2} \end{array}$$
(9)$$\begin{array}{l} Py_{max}=\frac{H_{List}.getmax(AsRAM)}{VM_{List}.getmin(AsRAM)};\;Py_{min}=\frac{H_{List}.getmin(AsRAM)}{VM_{List}.getmax(AsRAM)}\\ Py=\frac{Py_{min}+{Py}_{max}}{2} \end{array}$$
(10)$$\begin{array}{l} Pz_{max}=\frac{H_{List}.getmax(BU)}{VM_{List}.getmin(BU)};\;Pz_{min}=\frac{H_{List}.getmin(BU)}{VM_{List}.getmax(BU)}\\ Pz=\frac{Px_{min}+{Px}_{max}}{2} \end{array}$$
$P=\frac{{?}_{i=x}^{z}P_{i}}{3}$ // Calculating the average of all *p* values.
Initialize Centroids to empty
*1st centroid [1, 1] = *$VM_{List}.getavg\left(CPU\right)$*// Average CPU from VM list**1st centroid [1, 2] = *$VM_{List}.getavg(RAM)$*// Average RAM from VM list**1st centroid [1, 3] = *$VM_{List}.getavg\left(BU\right)$*// Average BU from VM list**Append 1st centroid to Centroids*h1=List.Centroid(CPU,RAM,BU)*// 1st centroid attributes*h2=List.VM(CPU,RAM,BU)*// VMs attributes*(11)$$Cos_{sim}=\frac{\sum_{i=1}^{n}h1_{n}\times h2_{n}}{\sqrt{\sum_{k=1}^{n}h1_{k}^{2}}\times \sqrt{\sum_{k=1}^{n}h2_{k}^{2}}}$$iCalculate the rest (P−1) of the centroids$VM_{remaining}=VM_{List}$*CS = []; // Initialize cosine similarity to 0*if P>0$CS\left[i\right]=GetCoSim(VM_{remaining}\left[i\right].getattributes(),$ *1st* centroid)$CS_{index}=Findmin_{index}(CS)$$Next_{Centroid}\left(1,1\right)=VM_{Lis}.CS_{index}.get(CPU)$$Next_{Centroid}\left(1,1\right)=VM_{Lis}.CS_{index}.get(RAM)$$Next_{Centroid}\left(1,1\right)=VM_{Lis}.CS_{index}.get(BU)$P=P−1$VM_{remaining}.Remove[CS_{index}]$End if$End_{for}$$For_{each}\;vm\;in\;VM_{remaining}$Simv *= [ ] // Initialize simv to empty*$For_{each}\;cent\;in\;Centroid$Simv.append(GetCoSim(VM,cent)$End_{For}$$Find\;\underset{\mathrm{index}}{\min}v=min\;(Simv)$$Allocate\;vm\;to\;Centroid.\underset{\mathrm{index}}{\min}\;v$
// Prioritization of the created clusters
$avg_{similarity}\;=\;[]$$Calculate\;avgSimilarity\;and\;Append\;to\;avg_{similarity}$$Append\;avgSimilarty\;to\;avg_{similarity}$$End_{for}$$Priortized\;=\;Sort.ascending(avg_{similarity})$Return Priortized



The proposed work modifies the current state of art with the alterations in the calculation of the similarity indexes. With the alteration in the similarity evaluation and added parameter of BU, the proposed work results in better power efficiency and violation ratios. The results are illustrated in the next section.

## 5. Results and Discussion

In order to compare the proposed work with existing algorithm architectures, two parameters, namely the overall consumed power and SLA-V, and in order to compute the SLA-V, again power consumption has been utilized along with a total number of migrations per PM. The power consumption has been evaluated in watts whereas the SLA-V is unit less and is defined in Table 1.

The analysis has been done taking the total VM in the incremental ratio of 100 VMs per 10 PMs, viz. the VMs range is 100–1000 VMs whereas the PMs range is 10–100 PMs. The power consumption is in watts and has been evaluated using Equation (3). As the total number of VMs increases, the average consumption also increases gradually. It is not necessary that if the load increases and the power consumption should also increase until and unless the variation of the load is significantly high. In most of the observed cases, the average power consumption increases. The average consumption of power can be illustrated using Figure 3 as follows.

In order to evaluate one scenario, the proposed work and other compared algorithms/techniques have been evaluated for 10,000 simulations. The average consumed power in the case of the proposed work scenario is improved by 3.8% as compared to [1], 3.14% as compared to [8], and 3.92% as compared to [28]. The effect of migration is evident when the total number of VMs increases after 200 VMs. When the load increases over the system, the system with sustainable allocation architecture will be more efficient in terms of holding the VMs at the right PM and discarding the VMs that are not required at the moment. As shown in Figure 3, the proposed work demonstrates minimum power consumption with the least number of migrations, the performance is evaluated to be improved overall. This is due to the additional evaluation measure added to the distance calculation and with additional input parameters, viz. BU in the assembly of the VMs. The proposed work evaluates the overall SLA-V based on both the parameters, including the total number of migrations and total power consumption, as follows.

As the proposed work balances both the total number of migrations and the power consumption parallelly, the overall SLA-V for the proposed work is quite economical in relation to other state-of-art techniques, as shown in Figure 4. The average overall SLA-V for the proposed work is 0.201 in all the scenarios, whereas [1,8,28] demonstrate 0.241, 0.2314, and 0.2712 for the same load conditions.

## 6. Conclusions

This paper presented an advanced algorithm architecture that was inspired by Nashaat et al. [8] architecture. The proposed work modified the current state of art technique by adding two utility parameters to the system, namely the bandwidth utilization and the cosine similarity to enhance the overall consumed power in order to allocate and migrate the VMs from the Nashaat et al. presented SESA algorithm, which is enhanced by utilizing more similarity indexes [8]. This research work is inspired by the research done by Beloglazov et al. [1], Kansal et al. [6], Nashaat et al. [8] and Garg et al. [28]. Due to improved utility function, the proposed work outcasts [1,8,28] by 3.8% as compared to [1], 3.14% as compared to [8], and 3.92% as compared to [28]. The evaluation has also been made on the basis of the total number of migrations and as the VM selection is precise in the case of the proposed work; the VMs are migrated in less quantity maintaining the overall power consumption in the case of the proposed work. The overall SLA V in all the presented scenarios, viz. 10–100 PMs supplied for 100–1000 VMs, is the least, as compared to other state-of-art techniques.

## Figures and Tables

**Figure 1 sensors-23-06117-f001:**
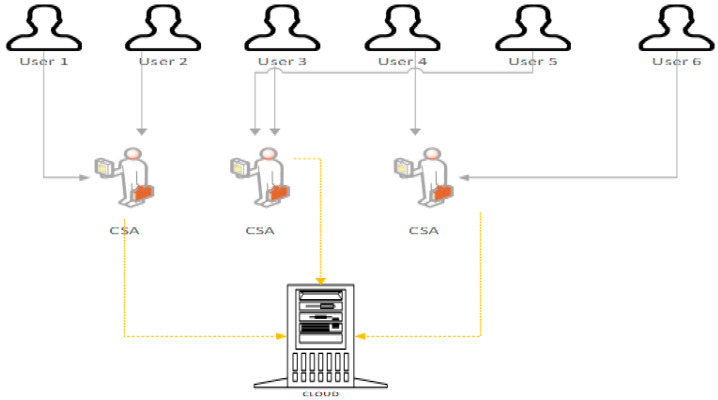
Green Cloud Components.

**Figure 2 sensors-23-06117-f002:**
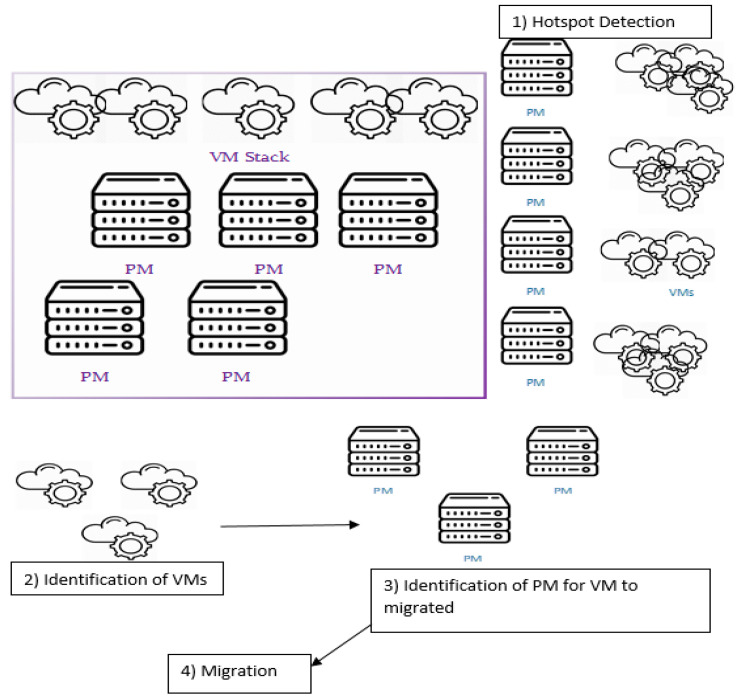
VM Allocation [1,8].

**Figure 3 sensors-23-06117-f003:**
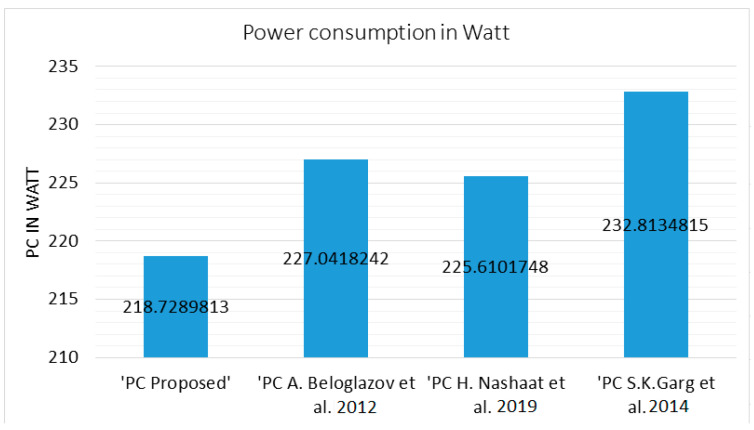
Average Power Consumption against [1,8,28].

**Figure 4 sensors-23-06117-f004:**
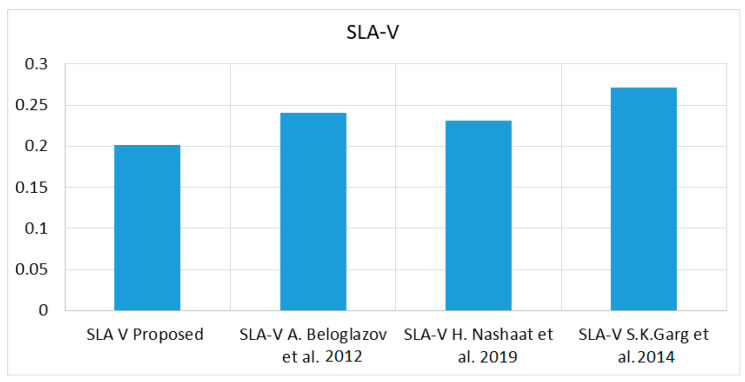
SLA-V against [1,8,28].

**Table 1 sensors-23-06117-t001:** (**a**) Power Consumption. (**b**) VM Migration Count.

(a)					
‘Total Number of VMs’	‘Total Number of PMs’	‘PC Proposed’	‘PC A. Beloglazov et al. [1]’	‘PC H. Nashaat et al. [8]’	‘PC S.K. Garg et al. [28]’
100	10	36.19172	39.48025	39.46517	37.844
200	20	82.25001	91.4687	83.53219	87.8643
300	30	123.8285	128.453	131.6822	126.29
400	40	163.2594	172.086	164.9016	167.7453
500	50	216.4734	218.5042	227.9728	219.2439
600	60	252.1217	254.6763	264.2169	259.984
700	70	251.7995	256.1971	255.911	278.066
800	80	316.4311	329.7573	335.92	331.6884
900	90	321.5957	337.6987	323.512	356.4329
1000	100	423.3387	442.0967	428.9879	462.976
(**b**)					
**‘Total Number of VMs’**	**‘Total Number of PMs’**	**‘Number of Migrations Proposed’**	**‘Number of Migrations A. Beloglazov et al. [1]** **’**	**‘Number of Migrations H. Nashaat et al. [8]** **’**	**‘Number of Migrations S.K. Garg et al. [28]** **’**
100	10	49	49	49	52
200	20	95	95	99	97
300	30	148	153	160	160
400	40	212	216	233	226
500	50	268	268	293	294
600	60	310	326	344	317
700	70	343	366	349	378
800	80	343	353	348	351
900	90	480	508	508	499
1000	100	467	481	490	472

## Data Availability

Data would be made available on request.

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
