# Peer review of "Algorithmic Approach to Virtual Machine Migration in Cloud Computing with Updated SESA Algorithm"

_sensors, 2023, doi:10.3390/s23136117_

Round 1

Reviewer 1 Report

1. In section 4.3, there is a reference to section 3.4, but this section is not included.

2.The last author's contribution was that there was a problem with the title layout and there was no alignment.

3. In the conclusion of the paper, the author can add some content that needs further research and improvement in the future.

Author Response

Comment 1: In section 4.3, there is a reference to section 3.4, but this section is not included.

Reply: The reference mentioned in the section 4.3 is for section 4.4(mistakenly written 3.4 in the text)

Once the VMs are selected, other than hotspot PMs, the rest PMs are listed down as per the required essentials and then the VMs are placed as shown in the following section 4.4. The listed PMs are now called available PMs.

Comment 2: The last author's contribution was that there was a problem with the title layout and there was no alignment.

Reply: In author’s Contributions, (S.J was mistakenly written);  the following authors contributed to this research work Amandeep Kaur(A.K) , Saurabh Kumar(S.K.), Deepali Gupta(D.G), Yasir Hamid(Y.H), Monia Hamdi(M.H), Amel Ksibi(A.K.I),  Hela Elmannai(H.E), Shilpa Saini(S.S);  Conceptualization, A.K. and D.G.; methodology, D.G; software, S.K, and Y.H; validation, D.G. and S.S.; formal analysis, A.K and S.K; investigation, S.S; resources, M.H and A.K.I; data curation, H.E. and A.K.I.; writing—original draft preparation, A.K and S.K; writing—review and editing, D.G and S.S.; visualization, Y.H and S.K; supervision, D.G; project administration, H.E , A.K.I and M.H.; funding acquisition, , H.E , A.K.I and M.H. All authors have read and agreed to the published version of the manuscript.

Comment 3: In the conclusion of the paper, the author can add some content that needs further research and improvement in the future.

Reply: We are thankful for this valuable comment. The conclusion has been improved in the revised manuscript. Kindly refer to section 6 conclusion of the revised manuscript for more details.

In addition, an integrated approach with swarm intelligence-based algorithm can be used in future for further development in this area in order to find the optimal VM al-location and migration and also minimize the problem of misallocation using minimization of migration which can reduce the significant amount of time and energy during the migration process while meeting the SLA constraints.

Reviewer 2 Report

This paper focused on the VM migration in cloud computing, for improving the power consumption and SLA violation. This is a research hotspot, and there are numerous related works published every year. There are many serious issues in the paper, to name a few,

1. The presentation of authors' work is terrible, making it hard to be understood.

2. The abstract should be improved by highlighting the contributions.

3. The background in Introduction section is too prolix. The innovations and contributions should be illustrated.

4. In related work section, the pros and cons of related works should be illustrated, instead of just “listing” them. The differences of their work from existing works should also be explained. Besides, new published works should be discussed. All related works are published before 2020 in this paper.

5. The presentation of the proposed algorithm should be improved to improve its readability.

6. New works should be used for the performance comparison.

Author Response

Comment 1: The presentation of authors' work is terrible, making it hard to be understood.

Reply: The We are thankful for this valuable comment. The presentation of the overall manuscript has been improved by improving the abstract, introduction, related work, methodology and also improved the readability of the proposed algorithm in the revised manuscript.

Comment 2: The abstract should be improved by highlighting the contributions.

Reply: The revised abstract shows the contribution of the present work.

Cloud Computing plays an important role in every IT sector. Several IT giants like Google, Microsoft, and Facebook as deploying their data centers around the world to provide computation and storage services. The customers either submit their job directly or take the help of the brokers for the submission of the jobs to the cloud centers. With an advancement in the technology, researchers came up with new terminologies and algorithmic architecture for the sustainability and reduction of power consumption which was ignored in the initial development of cloud services. This was due to the performance expectations from cloud servers as they were supposed to provide all the services through their service layers IaaS, PaaS, and SaaS. In this paper, an energy efficient framework has been designed for VM allocation and migration by meeting the Service Level Agreement (SLA). The proposed work employs an elastic scheduling-inspired Smart Elastic Scheduling Algorithm (SESA) to develop a more energy-efficient VM allocation and migration algorithm. In addition, a further enhancement in the proposed work has been done by using additional parameters such as cosine similarity and bandwidth utilization in order to improve the current performance in terms of QoS. The proposed work is evaluated for overall power consumption and SLA-violation (SLA-V) and is compared with existing state of art techniques. A Proposed algorithm is also presented to solve problems found during the survey such as optimal resource utilization, migration minimization.

Comment 3: The background in Introduction section is too prolix. The innovations and contributions should be illustrated.

Reply: The revised introduction is made more concise and shows the innovation and contribution of the present work.

Considering load as an effective parameter, Mann et al. presented a computation core-aware load-based MBFD algorithm [4]. The "de Castro et al." introduced the con-cept of Natural Computing [5], VM selection based on swarm intelligence(SI)[6] and hybrid SI [7] strategies has been discussed. The energy consumed during the migration is a prominent issue. The evaluation of "Consumed Energy (CE)" as shown in Equation (4) in which a source node is selected by the computation of the "Attraction Index (AI)," which is computed by.

In this work, the main issue of power consumption, SLA-V and optimal resource utilization has been addressed by allocating VM and migrating to the best available host in an energy efficient manner. It has been done by developing an energy efficient model using modified SESA algorithm and in order to improve the efficacy of the proposed model two additional parameters such as cosine similarity and bandwidth utilization has been employed.

Comment 4: In related work section, the pros and cons of related works should be illustrated, instead of just “listing” them. The differences of their work from existing works should also be explained. Besides, new published works should be discussed. All related works are published before 2020 in this paper.

Reply: The revised related work has been arranged in chronological form with pros and cons. Also added few latest works in the existing literature. Kindly refer literature survey section 2 in the paper.

Comment 5: The presentation of the proposed algorithm should be improved to improve its readability.

Reply: The proposed algorithm readability has been improved. Kindly refer section 4.4 and4.5.

Comment 6: New works should be used for the performance comparison.

Reply: The reference [35] has been added for the with the proposed technique. Kindly refer the table 1, table 2 and table 3 in the Section 5 result and discussion

Reviewer 3 Report

Based on the analysis of existing solutions for the migration of virtual machines in the cloud (which the authors analyze to identify weak points) in order to reduce energy consumption at the cloud level, the authors come up with a new solution for the allocation and migration of virtual machines in the cloud, which prove more efficient in terms of energy consumption.

The proposed approach uses an intelligent elastic planning algorithm (inspired from Naashat et al. architecture), and introduces 2 additional utility parameters to improve the current performance in terms of QoS: Bandwidth utilization and the Cosine similarity. In addition to this, the allocation is done based on the clusters' density only, which the authors intend to improve in their future work. 

Weakenesses:

- Ideas must be expressed more clearly and coherently.

- Line 50 must be deleted!

- I recommend a chronological reorganization of the literature survey section

- Section 4 presenting the methodology proposed for the research should be expressed more clearly. It should be rewritten for clarity.

- The conditions under which the results presented in the paper were obtained are not clear. Ask for clarification in this regard as well

Author Response

Responses to Reviewer: 3

Comment 1: Ideas must be expressed more clearly and coherently.

Reply: To provide more clarity to the overall concept of the paper. In abstract and introduction, contribution of the paper has been added. Methodology and result section also improved. Finally, future scope in the conclusion is added.

Comment 2: Line 50 must be deleted

Reply: Line 50 “Identify the applicable funding agency here. If none, delete this text box” has been deleted

Comment 3: I recommend a chronological reorganization of the literature survey section

Reply: The literature in chronological form with pro and cons has been added. Also latest literature work is included. Kindly refer section 2 literature survey

Comment 4: - Section 4 presenting the methodology proposed for the research should be expressed more clearly. It should be rewritten for clarity.

Reply: The methodology has been improved in order to provide more clear approach. Kindly refer the section 4.

Comment 5: The conditions under which the results presented in the paper were obtained are not clear.

Reply: The results were presented using Energy consumption (Watt) Vs Number of VMs; Number of migrations Vs Number of VMs; and SLA-Violations Vs Number of VMS; Kindly refer the Figure 2, Figure 3, and Figure 4 in the result and discussion, section 5

Round 2

Reviewer 2 Report

Image resolution should be improved. The presentation of algorithms should be improved. More new related works should be included.

Author Response

Reviewer Comments 

Author Responses

Image resolution should be improved

Figure 1 and figure 2 are improved

The presentation of algorithms should be improved

Presentation of Algorithm 1, algorithm 2, algorithm 3, and pseudo code is refined

More new related works should be included

A survey of 2022 papers is added in the literature section
